Evaluation of different Kabuli chickpea genotypes against Helicoverpa armigera (Hübner) (Lepidoptera: Noctuidae) in relation to biotic and abiotic factors

Muhammad Bilal Yousuf Hafiz 1 2
Yasin Muhammad 1 yasin_1876@yahoo.com
http://orcid.org/0000-0003-4650-0345 Ali Habib 3 habib.ali@kfueit.edu.pk
Naveed Khalid 4
Riaz Ammara 5
AlGarawi Amal Mohamed 6
Hatamleh Ashraf Atef 6
Shan Yunfeng 7
1 Department of Entomology, Faculty of Agriculture and Environment, Islamia University, Bahawalpur , Bahawalpur, Punjab , Pakistan
2 Department of Entomology, University of Agriculture Faisalabad , Faisalabad, Punjab , Pakistan
3 Department of Agricultural Engineering, Khwaja Fareed University of Engineering and Information Technology , Rahim Yar Khan , Pakistan
4 Department of Entomology, University of Agriculture Faisalabad, Depalpur Campus, Okara , Faisalabad , Pakistan
5 Department of Life Sciences, Khwaja Fareed University of Engineering and Information Technology , Rahim Yar Khan , Pakistan
6 Department of Botany and Microbiology, College of Science, King Saud University , Riyadh , Saudi Arabia
7 College of Mathematics and Computer Science, Guangxi Science and Technology Normal University , Laibin , China
Kumar Ravinder
Electronic publication date: 2024 Mar 13
Publication date: 2024
Volume: 12
Electronic Location ID: e16944
Received 2023 Oct 27; Accepted 2024 Jan 24
Copyright: © 2024 Muhammad Bilal Yousuf et al.
Copyright year: 2024
Copyright holder: Muhammad Bilal Yousuf et al.
License: This is an open access article distributed under the terms of the Creative Commons Attribution License, which permits unrestricted use, distribution, reproduction and adaptation in any medium and for any purpose provided that it is properly attributed. For attribution, the original author(s), title, publication source (PeerJ) and either DOI or URL of the article must be cited.
License URL: https://creativecommons.org/licenses/by/4.0/

Keywords: Chickpea, Physio-morphic characters, Trichomes, Genotypes, Chlorophyll contents, Abiotic factors, Genetic resilience

Funding: Guanxi Science and Technology Normal University GKS 20220901 This work was supported by a grant for high-level talents of Guanxi Science and Technology Normal University (GKS 20220901). The funders had no role in study design, data collection and analysis, decision to publish, or preparation of the manuscript.

==============================
Background

The chickpea pod borer Helicoverpa armigera (Hübner) is a significant insect pest of chickpea crops, causing substantial global losses.

Methods

Field experiments were conducted in Central Punjab, Pakistan, to investigate the impact of biotic and abiotic factors on pod borer population dynamics and infestation in nine kabuli chickpea genotypes during two cropping seasons (2020–2021 and 2021–2022). The crops were sown in November in both years, with row-to-row and plant-to-plant distances of 30 and 15 cm, respectively, following a randomized complete block design (RCBD).

Results

Results showed a significant difference among the tested genotypes in trichome density, pod wall thickness, and leaf chlorophyll contents. Significantly lower larval population (0.85 and 1.10 larvae per plant) and percent damage (10.65% and 14.25%) were observed in genotype Noor-2019 during 2020–2021 and 2021–2022, respectively. Pod trichome density, pod wall thickness, and chlorophyll content of leaves also showed significant variation among the tested genotypes. Pod trichome density and pod wall thickness correlated negatively with larval infestation, while chlorophyll content in leaves showed a positive correlation. Additionally, the larval population positively correlated with minimum and maximum temperatures, while relative humidity negatively correlated with the larval population. Study results explore natural enemies as potential biological control agents and reduce reliance on chemical pesticides.

Introduction

Chickpea, Cicer arietinum L. (Fabales: Fabaceae) is a widely cultivated grain legume crop in both tropical and temperate climates, commonly known as the “king of pulses” (Muehlbauer & Sarker, 2017; Ali, Aslam & Nadeem, 2022). Asia is the large producer of chickpea, accounting for 90% of the total production around the globe, followed by Africa, which accounts for 5.9% (Ahmed & Awan, 2013; Ali et al., 2021a). In Pakistan, C. arietinum is cultivated in rainfed and irrigated areas, accounting for 944 thousand hectares with an annual production of 438 thousand tons (Economic Survey of Pakistan, 2019–2020). Chickpea seeds are a good source of dietary fiber, protein, and carbohydrates and their leaves and dried stalks can be used in animal feed (Argaye, Keneni & Bayissa, 2021; Grasso et al., 2021; Khanzada et al., 2022; Maryam et al., 2023).

In Pakistan, the gram pod borer, Helicoverpa armigera (Hübner) (Lepidoptera; Noctuidae), is considered to be a notorious insect pest of chickpea crop that affects both quality and yield and causes substantial economic losses (Sarwar, Ahmad & Toufiq, 2009; Ali, Aslam & Nadeem, 2022). It is a polyphagous pest and attacks many other crops, including cotton, pigeon pea, maize, tomato, sunflower, sorghum, and a variety of vegetables and fruit crops and tree species (Devi, Sharma & Rao, 2011; Ali et al., 2021b). In severe cases, the pest can cause yield losses of up to 90%, depending on the insect density and susceptibility of the host crop (Mahmood et al., 2021; Shabir, Sarwar & Ali, 2023).

Insecticides have been extensively used to manage this pest on many crops. However, the adverse effects of insecticides, such as resistance development, hazardous effects on the environment and human and animal health, and disturb the natural balance between the beneficial agents (pathogens, parasitoids, and predators) and pest population in agro-ecosystem (Asif et al., 2018). This has led the researcher to find environmentally friendly alternatives that are safer for the environment and compatible with human health (Kranthi et al., 2002; Hanley et al., 2007; Singh, Sinha & Jamwal, 2010; Nawaz et al., 2021). Researchers have adopted various alternative techniques to mitigate the use of chemical insecticides. Among those, host plant resistance is a practical, economical, and environment-friendly pest control method that promotes the production of healthy products free of pesticide residues (Ali, Aslam & Nadeem, 2022). It must be considered an essential component of an integrated crop management system to control insect pests. The resistance mechanism in plants is either constitutive or induced and is categorized as antixenosis (non-preference), antibiosis, and tolerance (Painter, 1951). Antixenosis resistance mechanism deters the insect from colonization, feeding, movement, oviposition, and growth and development (Afzal et al., 2009; Suzana et al., 2015; Mamoon-ur-Rashid et al., 2022).

On the other hand, physiomorphic traits based on morphological characteristics of the plant, such as trichome density, pod wall thickness, chlorophyll contents, etc., attributes for antibiosis and antixenosis mechanisms of resistance against pod borer infestation (Altaf, Azizul & Prodhan, 2008; Sallmath et al., 2008). These characteristics can potentially reduce the visual appeal of the plant and serve as effective physical obstacles against pests. For instance, the presence of thick cell walls and plant tissue aids in the plant’s ability to withstand the damaging effects caused by chewing mouthparts of insects and prevents the penetration of an insect’s stylet or ovipositor. Similarly, some insects experience difficulty feeding on and ingesting plants with trichomes. They may also release sticky substances that can trap or inhibit the movement of small insects. Moreover, these traits are heritable within plants that reduce the pest population (Dogimont et al., 2010; Khan et al., 2021).

Likewise, abiotic stresses are the major contributors to pest control (Galav et al., 2018). These abiotic factors also play a fundamental role in changing the crop pest infestation such as temperature (14–45 °C), relative humidity (15–95%), and optimum and intermittent precipitation have been found to affect the population build-up, adult growth and maturity of female pod borer larvae (Basit et al., 2021; Karar et al., 2021; Hira et al., 2022). The utilisation of resistant varieties has emerged as a pivotal component in the triumph of numerous ongoing insect pest management initiatives. This approach has proven to be effective, feasible, economically viable, and environmentally friendly for pest management (Gemechu et al., 2012). If the farmers are provided with resistant varieties of chickpea, they will immediately accept them, as they do not want to invest more money in chemical pest control. Unfortunately, no sufficient information is available on chickpea genotypes for resistance to pod borer in Pakistan. Therefore, the present research was carried out to investigate the effect of physio-morphic characters and meteorological factors in nine Kabuli chickpea genotypes on pod borer population build-up and pod infestation under field conditions.

Materials and Methods

Experimental site

The field experiment was conducted at the agricultural farm area of the Entomological Research Institute, Ayub Agriculture Research Institute (AARI), Faisalabad, Pakistan. The study was conducted for two consecutive years, 2020–2021 and 2021–2022. Faisalabad is located 31° 25′ 7.3740″ N and 73° 4′ 44.7924″ E, and 186 m elevation. Faisalabad’s daily mean maximum and minimum temperatures are 45 and 19 °C, respectively. The soil of the experimental site was well-drained and loamy, with a significant proportion of silt and a pH of 8.2.

Land preparation and sowing

Nine commonly grown kabuli chickpea genotypes viz., K-01209, K-01211, K-01216, Noor-2019, K-01240, K-01241, K-01242, DG-2017 and K-01308 were sown on 10 November 2020 and 7 November 2021 with row-to-row and plant-to-plant distances of 30 and 15 cm respectively under a randomized complete block design (RCBD). The germplasm was obtained from the Pulses Research Institute, AARI. A pre-irrigated field for the experiment was prepared by ploughing and deep tilling with the help of a cultivator and a tractor MF-240 (model 2010). A non-experimental area (60 cm) was left on all the sides of the field, and 0.5 m broad strips separated each block. Chickpea seeds were sown in each block manually (chopa method) by labor. Standard cultural practices were adopted to maintain a good crop. Three irrigations were applied during the entire chickpea crop period, while all necessary agronomic practices were diligently executed to maintain a weed-free field. The fertilizer application consisted of Urea at a rate of 20 kg per hectare, diammonium phosphate at 100 kg per hectare, and potassium sulphate at a rate of 60 kg per hectare. Three parallel blocks were designated as three replicates, with each block consisting of nine plots measuring 2 × 3.5 m. These blocks were established during both the 2020/2021 and 2021/2022 cropping seasons.

Sampling

Crop growth rates were different for different genotypes, but phonological stages were recorded when 50% of the plants from each plot got 50% of branches, flowers, and physiological maturity (Mulwa, Kitonyo & Nderitu, 2023). All the genotypes were closely examined weekly from 25th February till 18th April. The number of pod borer larvae was recorded by randomly selecting five plants, whereas, in the case of percent pod damage, ten plants were selected randomly from each replication of each genotype, and the larval population and pod damage, along with the total number of pods on each plant was counted (Yadav et al., 2021). The sampling was done early in the morning when the temperature was low to avoid the pest becoming active. The larval population was counted on the lower, middle, and upper sections of the plant.

The following formula was used to find out the average population of pod borers (Ali, Aslam & Nadeem, 2022):

Averagepopulation=P1+P2+P3+P4+P55.

Percent pod damage was calculated by using the following formula suggested by Prakash & Arunkumar (2013).

Percentpoddamage=No.ofdamagedpodsperplantTotalno.ofpodsperplant×100.

Physiomorphic characteristics

Morphological traits of the crop, such as pod trichome density and pod wall thickness, were measured from ten randomly selected pods from each plot. Pod trichomes density (cm−2) was counted by observing the dorsal side of each pod under a binocular microscope at 100x magnification (Roshan & Raju, 2018). At physiological maturity, pod wall thickness (µm) was measured by the use of a vernier calliper in ten randomly selected pods per plot (Karthik & Vastrad, 2018). For measuring chlorophyll contents of leaves (mg ml−1), ten leaves were randomly collected from each replication of each genotype. Chlorophyll contents of the sampled leaves were determined according to Arnon’s (1949) method in the Plant Physiology Laboratory, AARI, Faisalabad.

Effect of meteorological factors on the incidence of pod borer

Weather data, including daily minimum and maximum temperature (°C) and relative humidity (%) during both growing seasons, was obtained from the Department of Crop Physiology, AARI’s weather station. The temperature and relative humidity throughout the experimental period were correlated with the pod borer population to check the response of these abiotic factors to the population dynamics of the pod borer.

Statistical analysis

Data regarding population dynamics and percent pod damage was analyzed using analysis of variance (ANOVA) to assess the experimental sources of variation using GenStat 15th Edition (Payne et al., 2011). Prior to analysis, data was tested for normality and conformed to the requirements of ANOVA. Means were compared and separated using Fisher’s least significant difference (LSD) at P ≤ 0.05 (Shabbir et al., 2014). Simple linear regression analysis explored relationships between abiotic factors and pest numbers. Linear regression slopes were tested for significant differences from zero by Sigma Plot version 10.0 (Kitonyo et al., 2018).

Results

Effect of physiomorphic characters on pod borer population and percentage pod damage

Pod borer population on different chickpea genotypes during 2020–2021

The ANOVA revealed significant variations (P ≤ 0.05) among the different genotypes during the observed weeks. This suggested that the genotypes had a significant impact on the larval population. Furthermore, the study found a significant increase in the larval population over time, specifically after 8 weeks during 2020–2021 (Table 1) due to a rise in temperature and a decrease in humidity. Moreover, genotype Noor-2019 exhibited relatively high resistance to the larvae, with an average of 0.85 larvae per plant. On the other hand, genotype DG-2017 was the most susceptible, with a significantly higher average population of 1.37 larvae per plant.

Table 1 Average (±SE) pod borer population on nine different chickpea genotypes for the year 2020–2021.

Genotype	Week 1	Week 2	Week 3	Week 4	Week 5	Week 6	Week 7	Week 8	Average	
K-01209	0.87 ± 0.05 bc	1.09 ± 0.08 bc	0.98 ± 0.08 bc	1.11 ± 0.07 bc	1.38 ± 0.14 b	1.26 ± 0.09 b	1.49 ± 0.14 c	1.62 ± 0.16 c	1.23 ± 0.10 bc	
K-01211	0.76 ± 0.03 de	0.99 ± 0.06 de	0.88 ± 0.06 de	1.01 ± 0.08 de	1.23 ± 0.10 cd	1.10 ± 0.10 cd	1.34 ± 0.11 de	1.46 ± 0.12 e	1.10 ± 0.11 de	
K-01216	0.63 ± 0.02 fg	0.86 ± 0.04 fg	0.75 ± 0.05 fg	0.91 ± 0.06 fg	1.08 ± 0.09 e	0.97 ± 0.08 ef	1.17 ± 0.09 f	1.32 ± 0.14 f	0.96 ± 0.09 fg	
Noor-2019	0.54 ± 0.03 h	0.74 ± 0.03 h	0.65 ± 0.05 h	0.81 ± 0.07 h	0.97 ± 0.09 f	0.85 ± 0.09 g	1.04 ± 0.10 g	1.20 ± 0.10 g	0.85 ± 0.10 h	
K-01240	0.80 ± 0.04 cd	1.04 ± 0.08 cd	0.94 ± 0.07 cd	1.06 ± 0.09 cd	1.29 ± 0.11 c	1.17 ± 0.12 c	1.40 ± 0.15 d	1.54 ± 0.13 d	1.15 ± 0.14 cd	
K-01241	0.59 ± 0.02 gh	0.80 ± 0.04 gh	0.70 ± 0.04 gh	0.86 ± 0.05 gh	1.02 ± 0.08 ef	0.92 ± 0.09 fg	1.10 ± 0.08 g	1.26 ± 0.11 g	0.90 ± 0.08 gh	
K-01242	0.70 ± 0.03 ef	0.93 ± 0.07 ef	0.82 ± 0.05 ef	0.96 ± 0.09 ef	1.16 ± 0.12 d	1.04 ± 0.13 de	1.28 ± 0.12 e	1.40 ± 0.16 e	1.04 ± 0.12 ef	
DG-2017	0.98 ± 0.06 a	1.22 ± 0.10 a	1.11 ± 0.09 a	1.26 ± 0.11 a	1.51 ± 0.16 a	1.40 ± 0.16 a	1.67 ± 0.15 a	1.78 ± 0.18 a	1.37 ± 0.15 a	
K-01308	0.92 ± 0.05 ab	1.14 ± 0.09 b	1.04 ± 0.07 ab	1.18 ± 0.09 b	1.44 ± 0.11 ab	1.34 ± 0.13 a	1.57 ± 0.18 b	1.71 ± 0.14 b	1.29 ± 0.11 ab	
F Statistic	32.10	38.30	37.59	40.49	57.78	57.25	78.89	99.55	50.27	
P value	≤0.05	≤0.05	≤0.05	≤0.05	≤0.05	≤0.05	≤0.05	≤0.05	≤0.05	
LSD at 0.05	0.0797	0.0781	0.0769	0.0710	0.0750	0.0755	0.0723	0.0603	0.0747	
%CVS	6.11	4.61	5.09	4.03	3.52	3.90	3.12	2.36	3.93	
Note:

Means sharing the same letters within each column are not significantly different at 5% level of significance.

Pod borer population on different chickpea genotypes during 2021–2022

Similarly, significant variation (P ≤ 0.05) in larval population among the examined genotypes was recorded during the second year (2021–2022). The larval population exhibited a positive correlation with the time (weeks), reaching its peak after 8 weeks (Table 2). The genotype Noor-2019 exhibited a comparatively higher resistance, averaging 1.10 larvae per plant. On the other hand, the genotype DG-2017 displayed the highest susceptibility, with a significantly greater population of pod borer, averaging 1.70 larvae per plant.

Table 2 Average (±SE) pod borer population on nine different chickpea genotypes for the year 2021–22. Means sharing the same letters within each column are not significantly different at a 5% level of significance.

Genotypes	Week 1	Week 2	Week 3	Week 4	Week 5	Week 6	Week 7	Week 8	Average	
K-01209	0.80 ± 0.07 bc	1.09 ± 0.11 c	1.44 ± 0.14 b	1.30 ± 0.11 c	1.60 ± 0.19 bc	1.88 ± 0.20 b	2.04 ± 0.18 b	2.15 ± 0.24 b	1.54 ± 0.18 c	
K-01211	0.69 ± 0.09 de	0.94 ± 0.09 e	1.27 ± 0.11 d	1.19 ± 0.09 d	1.46 ± 0.14 d	1.70 ± 0.18 d	1.86 ± 0.21 d	1.95 ± 0.19 d	1.38 ± 0.13 e	
K-01216	0.58 ± 0.07 fg	0.79 ± 0.07 f	1.14 ± 0.09 ef	1.07 ± 0.10 ef	1.30 ± 0.11 ef	1.54 ± 0.14 f	1.66 ± 0.18 f	1.74 ± 0.21 f	1.23 ± 0.10 g	
Noor-2019	0.48 ± 0.04 h	0.65 ± 0.05 g	1.03 ± 0.08 g	0.98 ± 0.13 g	1.17 ± 0.13 g	1.40 ± 0.11 g	1.51 ± 0.14 g	1.60 ± 0.17 g	1.10 ± 0.08 h	
K-01240	0.74 ± 0.07 cd	1.02 ± 0.09 d	1.36 ± 0.13 c	1.24 ± 0.15 d	1.52 ± 0.19 cd	1.79 ± 0.21 c	1.95 ± 0.23 c	2.04 ± 0.20 c	1.46 ± 0.19 d	
K-01241	0.52 ± 0.08 gh	0.74 ± 0.06 f	1.07 ± 0.11 fg	1.02 ± 0.09 fg	1.24 ± 0.10 fg	1.48 ± 0.14 f	1.59 ± 0.18 f	1.67 ± 0.16 fg	1.17 ± 0.13 gh	
K-01242	0.63 ± 0.05 ef	0.88 ± 0.09 e	1.21 ± 0.15 de	1.12 ± 0.10 e	1.36 ± 0.16 e	1.62 ± 0.18 e	1.74 ± 0.21 e	1.83 ± 0.21 e	1.30 ± 0.18 f	
DG-2017	0.92 ± 0.10 a	1.27 ± 0.13 a	1.61 ± 0.19 a	1.49 ± 0.16 a	1.76 ± 0.19 a	2.03 ± 0.25 a	2.21 ± 0.25 a	2.33 ± 0.28 a	1.70 ± 0.21 a	
K-01308	0.86 ± 0.09 ab	1.19 ± 0.10 b	1.53 ± 0.14 a	1.41 ± 0.13 b	1.67 ± 0.15 b	1.95 ± 0.19 b	2.10 ± 0.19 b	2.21 ± 0.23 b	1.61 ± 0.26 b	
F Statistic	37.78	82.97	54.54	62.83	58.16	78.49	88.27	104.73	72.44	
P value	≤0.05	≤0.05	≤0.05	≤0.05	≤0.05	≤0.05	≤0.05	≤0.05	≤0.05	
LSD at 0.05	0.0748	0.0688	0.0826	0.0661	0.0798	0.0742	0.0775	0.0748	0.0727	
%CVS	6.25	4.18	3.68	3.18	3.18	2.51	2.42	2.22	3.03	

Percent pod damage on different chickpea genotypes during 2020–2021

Percent pod damage was accessed for 8 weeks after an attack by pod borer on different chickpea genotypes during 2020–2021. The genotype DG-2017 was highly susceptible, with percent pod damage (22.90%). The genotype Noor-2019 was found to be the most resistant with less percent pod damage (10.65%), as shown in Table 3.

Table 3 Mean (%±SE) pod damage on nine different chickpea genotypes for the year 2020–2021.

Genotypes	Week 1	Week 2	Week 3	Week 4	Week 5	Week 6	Week 7	Week 8	Average	
K-01209	6.06 ± 0.60 abc	10.47 ± 0.79 abc	12.00 ± 1.12 abc	15.72 ± 1.14 b	21.22 ± 0.92	23.48 ± 1.23 bc	31.83 ± 1.36 bc	35.48 ± 1.35 b	19.53 ± 1.01 bc	
K-01211	4.40 ± 0.53 cde	8.32 ± 0.55 cde	9.86 ± 1.04 cde	12.78 ± 0.97 cd	19.64 ± 1.17 bc	21.11 ± 1.19 cd	27.86 ± 1.18 de	29.79 ± 1.21 d	16.72 ± 0.76 de	
K-01216	2.84 ± 0.57 e	7.13 ± 0.94 def	8.40 ± 1.21 def	10.94 ± 0.99 def	15.75 ± 0.95 de	18.23 ± 1.07 e	21.88 ± 1.12 f	23.42 ± 1.16 e	13.57 ± 0.67 fg	
Noor-2019	2.14 ± 0.65 e	5.06 ± 0.69 f	6.57 ± 0.74 f	8.90 ± 0.76 f	12.45 ± 0.78 f	14.28 ± 0.78 f	17.26 ± 0.82 g	18.53 ± 0.85 f	10.65 ± 0.71 h	
K-01240	5.18 ± 0.63 bcd	9.53 ± 0.98 bcd	10.99 ± 0.90 bcd	14.56 ± 1.08 bc	20.33 ± 1.24 b	21.95 ± 1.16 cd	28.65 ± 1.26 cd	33.13 ± 1.24 c	18.04 ± 0.93 cd	
K-01241	2.41 ± 0.38 e	6.37 ± 0.72 ef	7.68 ± 0.68 ef	9.81 ± 0.71 ef	14.09 ± 1.05 ef	15.45 ± 1.06 f	19.20 ± 0.98 fg	20.03 ± 1.08 f	11.88 ± 0.76 gh	
K-01242	3.72 ± 0.48 de	7.65 ± 0.85 def	9.18 ± 0.92 cdef	12.12 ± 0.93 cde	17.60 ± 1.17 cd	19.92 ± 1.35 de	25.28 ± 1.23 e	27.98 ± 1.53 d	15.43 ± 0.97 ef	
DG-2017	8.04 ± 0.92 a	12.92 ± 1.09 a	14.00 ± 1.23 a	19.49 ± 1.27 a	25.14 ± 1.34 a	28.37 ± 1.50 a	35.22 ± 1.49 a	40.02 ± 1.61 a	22.90 ± 1.14 a	
K-01308	7.23 ± 1.19 ab	11.76 ± 0.97 ab	13.21 ± 1.15 ab	16.90 ± 1.11 ab	23.90 ± 1.25 a	25.44 ± 1.42 b	34.49 ± 1.27 ab	39.21 ± 0.46 a	21.52 ± 1.04 ab	
F Statistic	7.63	8.49	6.72	15.82	28.70	30.27	36.42	135.53	31.84	
P value	≤0.05	≤0.05	≤0.05	≤0.05	≤0.05	≤0.05	≤0.05	≤0.05	≤0.05	
LSD at 0.05	2.2971	2.6563	2.9292	2.6296	2.4088	2.4804	3.2151	2.0412	2.2438	
%CVS	28.42	17.44	16.57	11.28	7.36	6.85	6.92	3.97	7.77	
Note:

Means sharing the same letters within each column are not significantly different at 5% level of significance.

Percent pod damage on different chickpea genotypes during 2021–2022

A recurring pattern of pod damage was consistently observed throughout the following year (2021–2022). The genotype DG-2017 displayed the greatest pod damage (26.94%). This was closely followed by the genotype K-01308, which exhibited (25.99%) pod damage. In contrast, genotype Noor-2019 exhibited resistance, with 14.25% pod damage (Table 4). Again, it has been observed that the genotype Noor-2019 demonstrated a heightened level of resistance against the pod borer, whereas the genotype DG-2017 was found to be highly susceptible.

Table 4 Mean (%±SE) pod damage on nine different chickpea genotypes for the year 2021–2022.

Genotypes	Week 1	Week 2	Week 3	Week 4	Week 5	Week 6	Week 7	Week 8	Average	
K-01209	4.24 ± 0.20 bc	9.57 ± 0.82 bc	18.80 ± 1.16 ab	21.40 ± 1.72 abc	26.63 ± 1.86 b	32.25 ± 1.71 bc	37.82 ± 1.98 ab	42.16 ± 2.13 ab	24.11 ± 1.44 bc	
K-01211	3.46 ± 0.17 d	7.89 ± 0.76 d	16.55 ± 1.03 bcd	18.53 ± 1.45 bcd	24.65 ± 1.74 bc	29.05 ± 1.82 cd	34.08 ± 1.75 cd	38.21 ± 1.97 bc	21.55 ± 1.26 cd	
K-01216	2.60 ± 0.13 e	6.97 ± 0.61 ef	13.46 ± 0.95 de	14.99 ± 1.18 def	18.73 ± 1.57 de	22.78 ± 1.50 e	26.41 ± 1.47 e	29.51 ± 1.63 d	16.93 ± 0.89 e	
Noor-2019	2.10 ± 0.05 f	5.88 ± 0.55 g	11.04 ± 0.88 e	12.27 ± 0.97 f	15.27 ± 1.09 e	19.77 ± 1.32 e	22.24 ± 1.52 f	25.45 ± 1.58 d	14.25 ± 0.87 e	
K-01240	4.03 ± 0.19 c	8.89 ± 0.79 c	17.80 ± 1.37 abc	20.12 ± 1.59 bc	25.72 ± 1.83 b	30.31 ± 1.76 cd	35.36 ± 1.89 bc	42.03 ± 1.97 ab	23.03 ± 1.53 c	
K-01241	2.53 ± 0.07 e	6.35 ± 0.69 fg	12.13 ± 0.94 e	13.35 ± 0.79 ef	17.39 ± 1.11 e	21.27 ± 1.12 e	24.72 ± 1.53 ef	27.47 ± 1.30 d	15.65 ± 0.91 e	
K-01242	2.92 ± 0.07 e	7.52 ± 0.57 de	15.60 ± 0.90 cd	17.30 ± 1.16 cde	21.96 ± 1.27 cd	27.11 ± 1.36 d	31.63 ± 1.79 d	35.37 ± 1.86 c	19.93 ± 1.20 d	
DG-2017	4.87 ± 0.24 a	10.95 ± 0.84 a	20.35 ± 1.65 a	24.50 ± 1.72 a	30.66 ± 1.75 a	37.15 ± 2.05 a	41.11 ± 2.11 a	45.91 ± 2.25 a	26.94 ± 1.47 a	
K-01308	4.55 ± 0.17 ab	10.41 ± 0.91 ab	19.87 ± 1.61 a	22.50 ± 1.64 ab	30.29 ± 1.84 a	34.39 ± 1.73 ab	40.67 ± 1.99 a	45.25 ± 2.08 a	25.99 ± 1.51 ab	
F Statistic	48.91	36.46	10.47	9.47	22.04	21.84	33.84	22.26	25.02	
P value	≤0.05	≤0.05	≤0.05	≤0.05	≤0.05	≤0.05	≤0.05	≤0.05	≤0.05	
LSD at 0.05	0.4242	0.8891	3.1281	4.1030	3.5249	3.8604	3.5665	4.9387	2.7306	
%CVS	7.04	6.21	11.17	12.93	8.67	7.90	6.31	7.75	7.54	
Note:

Means sharing the same letters within each column are not significantly different at a 5% level of significance.

Physiomorphic characters of different chickpea genotypes

Pod trichome density (2020–2021 and 2021–2022)

A significant difference in trichome density among various chickpea genotypes was observed during both years. In the tested genotypes, there was an observed inverse relationship between trichome density and pod damage. The data collected for the observed years indicated that genotype Noor-2019 exhibited the highest pod trichome density, measuring 352.89 and 344.64 trichomes cm−2, respectively. On the other hand, genotype DG-2017 exhibited the least pod trichome density, measured at 215.09 and 208.35 cm−2, respectively, compared to all other genotypes (Fig. 1).

Figure 1 Trichome density on pods of different chickpea genotypes during both observed years (2020–2021 & 2021–2022).

Means sharing the same letter on the bars are not significantly different at 5% level of significance.

Pod wall thickness (2020–2021 and 2021–2022)

A significant difference in pod wall thickness was observed among tested chickpea genotypes during both observed years. Maximum pod wall thickness was observed in genotype Noor-2019 (0.36 and 0.35 µm, respectively), while the minimum was recorded in DG-2017 (0.22 and 0.21 µm, respectively), as shown in Fig. 2.

Figure 2 Pod wall thickness of different chickpea genotypes during both observed years (2020–2021 & 2021–2022).

Means sharing the same letter on the bars are not significantly different at 5% level of significance.

Chlorophyll contents of leaves (2020–2021 and 2021–2022)

In relation to the chlorophyll contents in leaves, a significant difference was observed among various chickpea genotypes during both years. The genotype DG-2017 exhibited the highest chlorophyll contents, measuring 1.64 and 1.66 mg ml−1, respectively. These values were found to be significantly different from the chlorophyll contents observed in all other genotypes. On the other hand, significantly low chlorophyll contents were recorded in genotype Noor-2019, measuring at 0.75 and 0.81 mg ml−1, respectively (Fig. 3).

Figure 3 Leaves chlorophyll contents of different chickpea genotypes during both observed years (2020–2021 & 2021–2022).

Means sharing the same letter on the bars are not significantly different at 5% level of significance.

Meteorological factors

Temperature (2020–2021 and 2021–2022)

The mean minimum and maximum temperatures were low during the initial weeks of March, recorded as 12.12 and 29.48 °C, respectively. Subsequently, there was a gradual rise in temperatures, reaching 19.83 and 35.08 °C during the third week of April (Fig. 4A). In the second year of observation, the mean minimum and maximum temperatures were low during the start of March (13.08 and 27.05 °C, respectively). Afterward, these temperatures experienced a gradual rise, reaching 22.31 and 41.82 °C by the third week of April (Fig. 4B).

Figure 4 Graphical representation of data regarding temperature and relative humidity during 2020–2021 and 2021–2022.

(A and B) Temperature; (C and D) relative humidity.

Relative humidity (2020–2021 and 2021–2022)

In contrast, the mean relative humidity during the morning (8 a.m.) and evening (5 p.m.) hours reached its peak values (74.30% and 46.70%, respectively) during the initial week of March. As the temperature increased, there was a corresponding decrease in relative humidity, reaching values of 41.40% and 26.50% during the third week of April (Fig. 4C). In the following year, the mean relative humidity in the morning (8 a.m.) and evening (5 p.m.) reached at peak values (77.45% and 46.17%, respectively) during the initial week of March. Subsequently, these values declined to 40.6% and 19.50%, respectively, by the third week of April (Fig. 4D).

Influence of physio-morphic characters on pod borer infestation

Data revealed strong association between physiomorphic characters and pod borer infestation in both testing years. Pod trichome density was negatively (R2 ≥ 0.9915 and 0.9752, respectively) associated with percent pod damage in both observation years (Figs. 5A, 5B), whereas, genotypes with greater number of trichomes harbor less pest. The highest trichome density was recorded on pods of genotype Noor-2019, while the lowest was recorded on DG-2017. The genotypes with the highest pod trichome density showed less damage than those with the lowest, and vice versa. Likewise, a negative and highly significant correlation between pod wall thickness and percent pod damage (R2 ≥ 0.9788 and 0.9674, respectively) was observed on different chickpea genotypes during both observed years (Figs. 5C, 5D). Genotype with the maximum pod wall thickness was Noor-2019, and the minimum was recorded in DG-2017, which showed the highest pod damage. On the other hand, a positive and highly significant correlation (R2 ≥ 0.9860 and 0.9776, respectively) was observed between chlorophyll contents and pod damage during both years. The highest chlorophyll contents were detected in leaves of genotype DG-2017 and exhibited maximum pod damage, the lowest chlorophyll contents were recorded in the leaves of genotype Noor-2019, which showed minimum pod damage compared to all other tested genotypes (Figs. 5E, 5F).

Figure 5 Correlation between physiomorphic characters and percent pod damage during 2020–2021 and 2021–2022.

(A and B) Trichome density; (C and D) pod wall thickness; (E and F) leaf chlorophyll contents.

Correlation between abiotic factors and pod borer population

A simple correlation was worked out between meteorological factors and the incidence of pod borer during both observed years. The results revealed a positive and highly significant correlation (R2 ≥ 0.8139 and 0.9054, respectively) between minimum temperature and pod borer population for both years (Figs. 6A, 6C). Similarly, maximum temperature also showed a positive and highly significant correlation (R2 ≥ 0.9495 and 0.9847, respectively) with the pod borer population (Figs. 6B, 6D). The minimum pod borer population was recorded in the first week of March (avg. 0.75 and 0.69 per plant, respectively) with a maximum temperature range of 29.48 and 27.05 °C, respectively, while the maximum pod borer population was recorded in the third week of April (avg. 1.48 and 1.95 per plant, respectively) with a maximum temperature range of 35.08 and 40.66 °C for both years.

Figure 6 Correlation between Abiotic factors and average larval population during 2020–2021 and 2021–2022.

(A and C) Minimum temperature; (B and D) maximum temperature.

In the case of relative humidity, there was a strong but negative correlation (R2 ≥ 0.7728 and 0.9555, respectively) between morning relative humidity (8:00 a.m.) and pod borer population during both years (Figs. 7A, 7C). Similarly, the evening relative humidity (5:00 p.m.) was also negatively correlated (R2 ≥ 0.7356 and 0.8603, respectively) with the pod borer population (Figs. 7B, 7D). The results showed an increasing population trend with decreasing relative humidity and vice versa. The minimum pod borer population was recorded in the first week of March (avg. 0.75 and 0.69 per plant, respectively) when the morning relative humidity was highest (74.30% and 77.45%, respectively). The maximum pod borer population was recorded in the third week of April (avg. 1.48 and 1.95 per plant, respectively) when the morning relative humidity was lowest (41.40% and 40.66%, respectively).

Figure 7 Correlation between Abiotic factors and average larval population during 2020–2021 and 2021–2022.

(A and C) Relative humidity at 8:00 a.m. (B and D) Relative humidity at 5:00 p.m.

Discussion

Cultivation of resistant genotypes of chickpea is considered to be the safest method of insect pest control, and identifying such genotypes from the local germplasm is a key component of IPM programs for the sustainable production of chickpea (Saleem et al., 2022). In Pakistan, chickpea breeding programs have significantly improved the adaptation to diseases and abiotic stresses. Still, limited attention has been given to insect pests, which particularly cause substantial economic losses. We performed this study in order to identify the high- and low-population of chickpea pod borer on nine different kabuli chickpea genotypes characterized by physiomorphic characteristics and abiotic factors. The finding revealed a significant difference in the pod borer population and physiomorphic characters of various chickpea genotypes. Crop morphological characteristics have been found to impact pest populations by physically disrupting the processes of host selection, feeding, ingestion, digestion, mating, and oviposition, as demonstrated by Quandahor et al. (2019). These traits exhibit a synergistic effect, wherein they interact to either enhance or diminish pest infestation rates. The genotypes with high trichome density (Noor-2019) exhibited a strong negative association with pod borer damage compared to those with low trichome density (DG-2017).

In accordance with our study, Shanower, Yoshida & Peter, 1997 observed that high numbers of non-glandular trichomes in pods of Pigeon Pea minimize the larval damage caused by the pod borer. Likewise, evidence from previous studies also revealed that leaf pubescence negatively affects insect behavior (Amjad, Bashir & Afzal, 2009, Khuram et al., 2011; Rustamani et al., 2014; Shabbir et al., 2014; Bayoumy et al., 2017, Quandahor et al., 2019; Mulwa, Kitonyo & Nderitu, 2023). It might be attributed to the fact that insects experience difficulty in feeding and ingesting the plant or plant parts with trichomes; they may also release sticky substances that can trap or inhibit the movement of insects.

Physical barriers such as pod wall thickness also significantly alter the insect feeding behavior. In our case, pod wall thickness differed significantly among the tested genotypes. The pod wall in the genotype (Noor-2019) was thicker than that of the susceptible genotype (DG-2017). Thicker pod wall provided a mechanical barrier and correlated negatively with pod borer infestation. Mulwa, Kitonyo & Nderitu (2023) observed lower pod borer damage in green gram genotypes with thicker pod walls. Apart from the physical barrier, pods exude toxic metabolites that alter the larval feeding behavior (Sharma, Shankhdhar & Shankhdhar, 2015). Our findings also confirm the results of Karthik & Vastrad (2018), who reported the lower pod borer infestation in the genotypes with thicker pod wall. Jat et al. (2018) testified that chickpea genotypes with thick pod wall exhibited resistance towards pod borer damage than the other genotypes.

Among the ecological variables, the quality of the host plant is an important indicator for determining variation in insect herbivory (Espírito-Santo et al., 2007). In our case, considering the biochemical characteristics (chlorophyll contents) of leaves, maximum pod borer damage was observed in genotypes with high chlorophyll contents compared to genotypes with low chlorophyll contents. Haralu et al. (2018) observed higher numbers of eggs in the chickpea genotypes with higher chlorophyll contents than with lower chlorophyll contents. Similar findings were also reported by Bommesha et al. (2012), who found a substantial positive association between the total chlorophyll content of leaves and leaf roller population in pigeon pea. It is speculated that higher chlorophyll content contributes to the greater palatability of plant tissues to herbivores insects (Sousa-Souto et al., 2018). On the other hand, chlorophyll and nitrogen contents of the plant have a positive correlation (Shadchina & Dmitrieva, 1995), and insects prefer tissues with high nitrogen content. Hence, in addition to the foraging, nitrogen-rich sites on plants are preferred for feeding and oviposition (Eubanks & Styrsky, 2005; Coelho, Veiga & Torres, 2009; Madritch & Lindroth, 2015). In agreement with previous studies, our study also demonstrated that increased chlorophyll content was positively associated with pod borer damage.

Abiotic factors negatively affect plant growth and production, leading to yield losses (Arun & Venkateswarlu, 2011; Ye et al., 2017). Furthermore, the cellular-level reaction of plants to abiotic stress is frequently interconnected, resulting in molecular, biochemical, physiological, and morphological alterations that impact plant growth, development, and productivity (Ahmad & Prasad, 2011; Nair et al., 2019). We observed a positive correlation between temperature (minimum and maximum) and larval population; the highest larval population was recorded during April in both years (2020–2021 and 2021–2022). The larval population started to build up in March and increased gradually with the increase in temperature. Contrarily, relative humidity (minimum and maximum) negatively correlated with the pod borer population. The findings were in accordance with Pal, Banerjee & Samanta, 2020, who found a positive correlation between temperature and pod borer population and a negative with the relative humidity. Similar findings were also reported by Kumar, Tripathi & Chandra, 2019 with pod borer population concerning abiotic parameters. The highest pod damage and larval population at high temperatures are attributed to the fast larval development. It has been observed in various insect species that an elevation in temperature leads to a decrease in the duration of the larval stage, which can be attributed to heightened metabolic rates and enhanced feeding activities (Johnston & Bennett, 2008; Srivastava & Omkar, 2003; Pervez, 2004; Tamiru, 2021).

The current findings are also consistent with those of Roshan & Raju (2018a), who found a positive correlation between the population of pod borers and the number of sunshine hours during 2015–2016 and 2016–2017. The correlation between maximum and minimum temperature and larval population during both years of experimentation was positive but not statistically significant. Further, rainfall, morning relative humidity, and evening relative humidity were observed to be negative and non-significantly correlated with pod borer population. In the rundown, the use of varietal screening in integrated pest management is crucial. This strategy eliminates the need for chemical pesticides by using the natural genetic variation in plants, which lowers costs, has a negligible negative impact on the environment, and increases sustainability. In addition to physiomorphic characteristics and abiotic factors, biochemical substances, such as semiochemicals and plant secondary metabolites like phenolic compounds, may have played a substantial role in determining pest preference among the green gram varieties under evaluation. This aspect warrants further investigation to gain a deeper understanding.

Conclusion

Host plant resistance offers a highly effective and promising approach to control pod borer populations in chickpea cropping systems. Our study observed variations in comparative resistance to pod borer populations among the tested genotypes influenced by plant physio-morphic characteristics, and abiotic factors. Multiple stepwise regression analyses suggest that, among the abiotic factors, maximum temperature plays a pivotal role in influencing gram pod borer population dynamics in chickpeas. Furthermore, our findings indicate a correlation between plant physio-morphic characteristics and average pod damage, with pod trichome density and pod wall thickness showing negative correlations with average pod borer damage. Contrarily leaf chlorophyll content exhibited a positive correlation with pod borer population. Consequently, pest-resistant genotypes can be acclimated to their respective environments, providing targeted, long-term protection while preserving beneficial organisms. In addition, these traits must be deployed in chickpea breeding program and align with contemporary ecological and consumer demands for safer and more sustainable agricultural practices. Therefore, we recommend that farmers utilize approved resistant varieties and that these traits be integrated into future breeding programs.

Supplemental Information

Supplemental Information 1 Raw data.

Additional Information and Declarations

Competing Interests

Author Contributions

Data Availability

Dr Habib Ali is an academic editor for PeerJ.

Hafiz Muhammad Bilal Yousuf conceived and designed the experiments, performed the experiments, prepared figures and/or tables, authored or reviewed drafts of the article, and approved the final draft.

Muhammad Yasin conceived and designed the experiments, performed the experiments, prepared figures and/or tables, authored or reviewed drafts of the article, and approved the final draft.

Habib Ali analyzed the data, prepared figures and/or tables, and approved the final draft.

Khalid Naveed analyzed the data, prepared figures and/or tables, and approved the final draft.

Ammara Riaz analyzed the data, prepared figures and/or tables, and approved the final draft.

Amal Mohamed AlGarawi analyzed the data, prepared figures and/or tables, and approved the final draft.

Ashraf Atef Hatamleh analyzed the data, prepared figures and/or tables, authored or reviewed drafts of the article, and approved the final draft.

Yunfeng Shan conceived and designed the experiments, performed the experiments, prepared figures and/or tables, authored or reviewed drafts of the article, and approved the final draft.

The following information was supplied regarding data availability:

The raw data is available in the Supplemental File.

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
