# Peer review of "Evaluation of different Kabuli chickpea genotypes against Helicoverpa armigera (Hübner) (Lepidoptera: Noctuidae) in relation to biotic and abiotic factors"

_PeerJ, doi:10.7717/peerj.16944_

## Round 0.1 · original submission · Major Revisions

The authors are requested to address all comments raised by the reviewers.

**Language Note:** The review process has identified that the English language must be improved. PeerJ can provide language editing services - please contact us at copyediting@peerj.com for pricing (be sure to provide your manuscript number and title). Alternatively, you should make your own arrangements to improve the language quality and provide details in your response letter. – PeerJ Staff

Reviewer 1 ·

Basic reporting

This study holds significance in the sustainable management of chickpea crops against a crucial insect pest; however, the current presentation is challenging to review. Kindly enhance the language of the manuscript thoroughly. Additionally, I have some suggestions to improve the content of this work.
Line 23: it is not clear biotic factors (physio-morphic characters)
Abstract: Please include statistically significant key results in the abstract

Experimental design

Introduction: Please improve the content and cite the references appropriately.
Is this the standard method to find the average population of pod borers. Give references.
Correlation data should be presented in a better manner.
Improve the bar graph to make it publishable.
Check the units in trichome density.

Validity of the findings

Findings are valid just minor corrections are required.

·

Basic reporting

The manuscript investigates the impact of biotic and abiotic factors on pod borer infestation in nine kabuli chickpea genotypes. The study was conducted over two cropping seasons in Central Punjab, Pakistan. The study addresses an important agricultural issue, the impact of Helicoverpa armigera on chickpea crops, which is of global significance. The study employs a comprehensive approach, considering both physiomorphic characteristics of the genotypes and abiotic factors like temperature and humidity. The use of statistical methods for data analysis, including ANOVA and correlation analysis enhances the rigor of the study. The study suggests exploring natural enemies as potential biological control agents, promoting environmentally friendly alternatives to chemical pesticides.

Experimental design

The manuscript clearly states its objective and provides a detailed description of the methods used in the field experiments. The experimental design, including the genotypes studied, the location, and the parameters measured, is well-described.

Validity of the findings

The results section effectively communicates key findings, such as differences in larval population and pod damage among genotypes.
Physiomorphic traits (pod trichome density, pod wall thickness, chlorophyll content) are well-documented, supporting potential correlations with pod borer infestation.

Additional comments

The manuscript lacks a clear structure. It would benefit from a more organized and logical flow of information, especially in the introduction and methods sections.
The background section, though informative, is extensive and could be condensed for better readability. The connection between the background and the specific research question could be more explicitly stated. The authors are suggested to read and cite the following latest articles to strengthen the manuscript
Ali, Q, Aslam A, Nadeem I. 2022. Genotypical variations and association between gram pod borer (Helicoverpa armigera) and physio-morphological traits in gram (Cicer arietinum L.). Plant Protection 6(2): 85-90.
Khanzada N, Iqbal O, Baby A, Zaman S, Tofique M, Rajput A. 2022. Evaluation of chickpea germplasm for relative resistance or susceptibility against Fusarium wilt and ascochyta blight under field conditions. Plant Protection 6(2): 121-13.

The presentation of data in the results section could be improved.
The conclusion is brief and does not adequately summarize the key findings or discuss their implications. A more comprehensive conclusion is needed.
The manuscript contains several language issues, including grammatical errors, awkward phrasing, and inconsistent language usage. A thorough proofreading and editing process is necessary to improve the overall clarity and fluency of the manuscript.
In summary, the manuscript addresses an important agricultural issue but requires substantial improvements in language quality, organization, and data presentation to enhance its suitability for publication.

Reviewer 3 ·

Basic reporting

Line 52-55 it's not clear if the uses of chickpea cultivation mentioned there are just about Pakistan or if they apply worldwide

Figures 6 and 7, it would be helpful if the graphs could use the same scale consistently. This would enhance the clarity of the trend line and make it easier to observe and compare patterns across the graphs.

Experimental design

no comment

Validity of the findings

It would be valuable to discuss whether there were notable differences between the two years assessed. The figures indicate slight changes between the years 2020-21 and 2021-22. Were there any distinctive factors contributing to the variations in results between these two years?

A more detailed exploration of the DG-2017 variety would be valuable, especially in understanding the factors that contribute to its comparatively inferior results compared to Noor-2019. Additionally, an explanation regarding the criteria used to select these nine varieties could provide readers with insights. Is there a shared characteristic or specific reason behind choosing these particular varieties?

---

## Round 0.2 · accepted · Accept

The revised manuscript is acceptable in its current state.

Reviewer 1 ·

Basic reporting

The manuscript is acceptable as it has worth sufficient data to support the farmer's needs.

Experimental design

Well-conducted experiments and the revised manuscript is looking good.

Validity of the findings

NA

Additional comments

NA

·

Basic reporting

Ok

Experimental design

Ok

Validity of the findings

Ok

Additional comments

The updated version of the manuscript titled “Evaluation of deferent Kabuli chickpea genotypes against Helicoverpa armigera (Hübner) (Lepidoptera: Noctuidae) in relation to biotic and abiotic factors” has undergone another rigorous review, during which the changes implemented following the previous review were critically examined.
In the resubmitted manuscript, the authors have made significant revisions to address the concerns raised in the earlier review. Furthermore, they have provided well-justified responses to various queries that were raised.
Through the process of addressing these queries and incorporating the suggested amendments, the manuscript has undergone substantial improvement and enhancement. The only remaining issues are minor grammatical errors, which can easily be rectified during the proofreading of the final galley if the manuscript is accepted.